# Unleashing the Power of Knowledge Graph for Recommendation via Invariant Learning

## ABSTRACT

Knowledge graph (KG) demonstrates substantial potential for enhancing the performance of recommender systems. Due to its rich semantic content and associations among interactive entities, it can effectively alleviate inherent limitations in collaborative filtering (CF), such as data sparsity or cold-start issues. However, most existing knowledge-aware recommendation models indiscriminately aggregate all information in KG, without considering information specifically relevant to the recommendation task. Such indiscriminate aggregation could introduce additional noisy knowledge into representation learning, which can distort the understanding of users' genuine preferences, thereby sacrificing the recommendation quality. In this paper, we introduce the principle of invariance to the knowledge-aware recommendation, culminating in our Knowledge Graph Invariant Learning (KGIL) framework. It aims to discern and harness the task-relevant knowledge connections within KG to enhance the recommendation models. Specifically, we employ multiple environment generators to simulate diverse noisy KG-environments. Then we devise a novel attention learning mechanism for KG and user-item interaction graph, aiming to learn environment-invariant subgraphs. Leveraging an adversarial optimization strategy, we enhance the diversity of the environments, meanwhile, promote invariant representation learning across environments. We conduct extensive experiments on three datasets and compare KGIL with state-of-the-art methods. The experimental results further demonstrate the superiority of our approach.

## KEYWORDS

Knowledge-aware Recommendation, Invariant Learning

## 1 INTRODUCTION

In the age of the Internet, information overload has intensified, making recommendation systems increasingly vital. They are now ubiquitously integrated into e-commerce platforms, search engines, and social media sites. The traditional recommendation model, exemplified by collaborative filtering (CF) [14, 15, 21, 38], operates on the premise that users with similar interaction behaviors tend to have analogous item preferences. While CF-based models have seen substantial success across various recommendation scenarios, their reliance on historical interaction data leads to challenges such as data sparsity and cold-start issues. Recently, the integration of

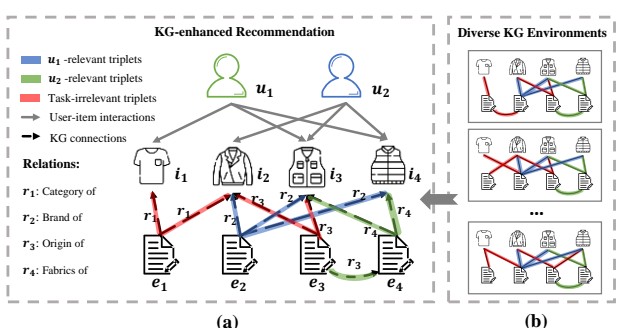

**Figure 1: (a) An intuitive example in an e-commerce recommendation scenario. $u_1$ and $u_2$ have different preferences, which can be reflected by their respective relevant triplets. (b) Diverse KG environments contain a variety of potentially noisy KG connections. Cross-environment invariant learning helps distinguish task-relevant connections.**

side information from knowledge graphs (KGs) has introduced an innovative dimension to recommendation models [2, 25], termed knowledge-aware recommendation. KGs encapsulate vast amounts of semantic knowledge and factual associations related to items. They not only offer solutions to the aforementioned challenges but also hold the promise of enhancing the robustness and explainability of recommendation systems [10, 32, 37, 41].

Existing knowledge-aware recommendation studies mainly fall into the following three categories: 1) Embedding-based methods [2, 34, 51]: These efforts primarily merge transition-based knowledge graph embeddings into item embeddings [22], offering a more comprehensive view of user and item modeling. 2) Path-based methods [16, 17, 40]: By capitalizing on meta-paths bridging users and items, these methods harness the inherent connectivity between users and items to a greater degree. 3) GNN-based methods [35, 37, 39, 43]: Drawing inspiration from the successes of graph neural networks (GNNs), they facilitate end-to-end aggregation of high-order information in KGs. This is achieved by crafting knowledge-aware aggregations, leading to enhanced user and item representation learning.

While the above studies have demonstrated efficacy in certain contexts, they frequently aggregate all the information present in given KGs without discerning its relevance to the ensuing recommendation tasks. As a consequence, the performance of the knowledge-aware recommender becomes heavily contingent on the quality of input KGs. Unfortunately, KGs often encompass a plethora of extraneous information, attributable to their long-tailed entity distributions or knowledge connections unrelated to the subject matter between items and entities. This redundancy often compromises the efficiency of KG-enhanced recommendation models, even sacrificing the recommendation quality. For instance, Figure 1 (a) shows a KG-enhanced recommendation scenario. $u_1$ and $u_2$ have purchased four pieces of clothing $i_1$, $i_2$, $i_3$, and $i_4$. $u_1$

interacts with $i_3$ and $i_4$ because of the preference of the $e_2$ brand, while $u_2$ interacts with $i_3$ and $i_4$ because of the focus on the clothing fabrics $e_4$. Hence, there exist informative paths, such as the path $u_1 \rightarrow i_3 \xrightarrow{r_4} e_4 \xrightarrow{r_4} i_4$ for $u_1$, and the path $u_2 \rightarrow i_2 \xrightarrow{r_2} e_2 \xrightarrow{r_2} i_3, i_4$ for $u_2$. These task-relevant connections enhance the capabilities of the recommendation model. In contrast, there also exist task-irrelevant noise connections in KG, such as the path $u_2 \rightarrow i_3 \xrightarrow{r_3} e_3$ and $u_2 \rightarrow i_2 \xrightarrow{r_1} e_1 \xrightarrow{r_1} i_1$. Indiscriminately absorbing the noise information will largely reduce the effectiveness of KG for recommendation. To address these issues, existing endeavors adopt contrastive learning to enhance knowledge semantics. For instance, [49, 54] employ data augmentation strategies to improve representation quality; [48, 53] implement contrastive learning to identify informative connections. Hence, these efforts filter out noisy knowledge by encouraging the model to learn representations that are invariant across views or environments.

While the aforementioned methods offer some merit, we contend that they do not sufficiently consider the diversity of environments inherent to KGs. Given the inherent variety of noise within KGs, confining the model to learn invariant representations in a limited set of KG-environments compromises its capacity to discern task-relevant knowledge. For illustration, Figure 1 (b) showcases KG data in multiple noisy environments that retain task-related knowledge connections. Intuitively, if a model is exposed to KG data across a broad spectrum of environments, it will inherently prioritize and assimilate from those connections stable across these noisy environments. Consequently, we posit that KG-enhanced recommendation models should simultaneously consider environmental diversity and cross-environmental invariance. Environmental diversity facilitates the model's exploration across a spectrum of noisy environments, while cross-environmental invariance enables it to discern disparities between these environments, directing its attention to pertinent task-related knowledge. This naturally prompts two challenges: How can we generate diverse KG-environments, and how can we ensure cross-environment invariant learning in knowledge-aware recommendations?

In this study, we introduce a novel framework, Knowledge Graph Invariant Learning (KGIL), designed to bolster the efficacy of KG for recommendations. Its conceptual foundation is rooted in invariant learning [3, 7, 19, 27], leveraging the principles of sufficiency and invariance to identify task-relevant knowledge connections from KGs, thereby enhancing the recommendation. Specifically, to generate KGs under diverse noisy environments, we employ multiple independent environment generators to simulate potentially noisy KG connections. Then we design invariant attention mechanisms within both the KG and interaction graph and encourage the model to capture attentive subgraphs. Finally, we adopt an adversarial optimization strategy to improve the diversity of environments and the invariance of representation learning.

In essence, our main contributions can be summarized as:

- We argue that encouraging the model to perform invariant learning across diverse noisy environments can effectively capture task-relevant knowledge in KG, which is beneficial to achieve robust and accurate recommendation models.
- We propose a novel framework KGIL. It can generate diverse noisy environments from KG and encourage the model to learn

representations of users and items that are invariant across these environments. Hence, it can effectively distill the task-relevant knowledge while filtering out the noisy connections.
- We conduct experiments on three KG recommendation benchmark datasets and compare them with existing state-of-the-art methods. Extensive experimental results and in-depth analyses demonstrate the effectiveness of our proposed KGIL.

## 2 PRELIMINARIES

In this section, we present the primary notations and definitions utilized in this paper. Subsequently, we offer a formal description of the knowledge-aware recommendation task.

### 2.1 Notations and Definitions

In a standard recommendation context, we have a set of users, denoted as $\mathcal{U}$, and a set of items, represented as $\mathcal{I}$. Let $u \in \mathcal{U}$ and $i \in \mathcal{I}$ be specific instances of a user and an item, respectively. The cardinalities of these sets are given by $|\mathcal{U}|$ for users and $|\mathcal{I}|$ for items. The interaction matrix, based on historical data such as clicks, purchases, or ratings, can be constructed as $\mathcal{Y} \in \mathbb{R}^{|\mathcal{U}| \times |\mathcal{I}|}$. An element $y_{ui}$ in this matrix indicates that user $u$ has previously interacted with item $i$. If no interaction took place, then $y_{ui} = 0$. For notation consistency, bold font (*e.g.*, $\mathbf{u}$, $\mathbf{i}$) indicates random variables, while italic font (*e.g.*, $u, i$) signifies their specific instances.

**User-Item Interaction Graph (IG).** From the interaction matrix $\mathcal{Y}$, we construct the user-item interaction bipartite graph, denoted as $\mathcal{G}_b = \{\mathcal{V}_b, \mathcal{E}_b\}$. The node set $\mathcal{V}_b = \mathcal{U} \cup \mathcal{I}$ includes both user and item nodes. The edge set, $\mathcal{E}_b = \{(u, i)|u \in \mathcal{U}, i \in \mathcal{I}, y_{ui} = 1\}$, encompasses interactions between users and items.

**Knowledge Graph (KG).** Knowledge graphs represent structured data capturing real-world semantic facts, such as concepts, common sense, or relationships between attributes. They provide auxiliary information to recommendation models, enriching the context for interactive entities. The KG is denoted as a heterogeneous graph, $\mathcal{G}_k = \{(h, r, t)|h, t \in \mathcal{V}_k, r \in \mathcal{E}_k\}$, where each entity-relation-entity triplet $(h, r, t)$ defines the graph. Here, $h$ and $t$ stand for the head and tail of knowledge entities, while $r$ signifies the semantic relation between them. The item set $\mathcal{I}$ from IG data forms a proper subset of the entity set $\mathcal{V}_k$, *i.e.*, $\mathcal{I} \subseteq \mathcal{V}_k$. We refer to the entities in the KG corresponding to items as "item entities". Other entity nodes are termed "attribute entities".

### 2.2 Task Formulation

We delineate the task description for our knowledge-aware recommendation. It can be defined based on the following components:

- **Data:** The user-item IG data $\mathcal{G}_b = \{\mathcal{V}_b, \mathcal{E}_b\}$ and the KG data $\mathcal{G}_k = \{(h, r, t)|h, t \in \mathcal{V}_k, r \in \mathcal{E}_k\}$.
- **Model:** We aim to devise a function, $\hat{y}_{ui} = f(u, i|\mathcal{G}_b, \mathcal{G}_k)$, capable of predicting potential interactions between a user $u$ and an item $i$. Here, $\hat{y}_{ui}$ symbolizes the prediction outcome. If $\hat{y}_{ui} = 1$, it suggests that user $u$ is likely to interact with item $i$. Conversely, $\hat{y}_{ui} = 0$ denotes no potential interaction.

In practice, the construction of KGs is often distinct from the user-item data collection process [48, 49, 53], leading to the inclusion of extraneous information in the KGs that might not pertain directly to the recommendation task. In addition, some studies [53]

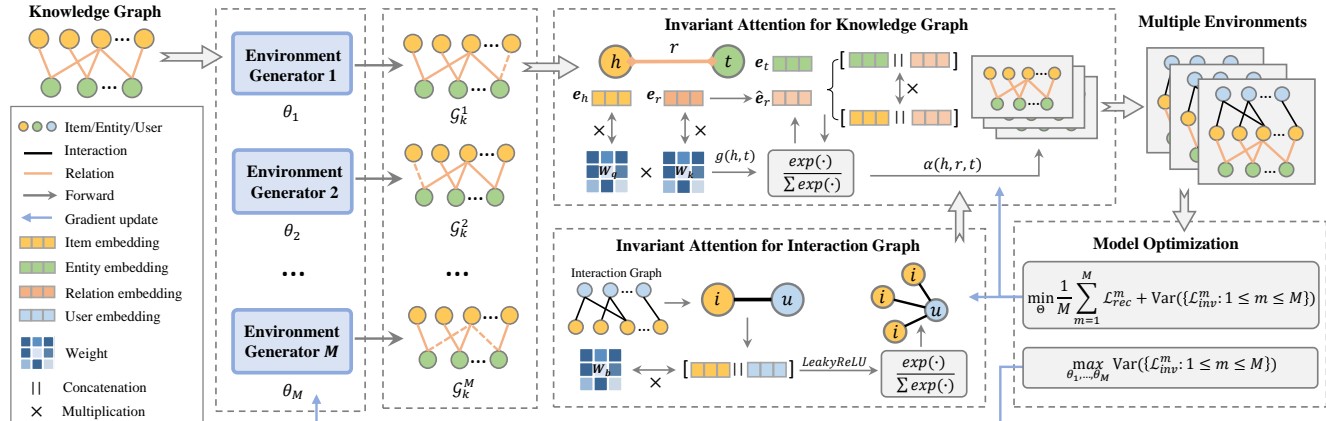

Figure 2: The overview of the proposed KGIL framework for knowledge-aware recommendation.

also point out that there also exist certain noisy interactions in IG. For instance, inadvertent clicks by users can introduce misleading signals for effective recommendation. The existence of noise information brings great challenges to the knowledge-aware recommendation. Owing to the unpredictability of these noises, we hypothesize that they are derived from diverse noisy environments. Let $\mathcal{S}$ represent the ensemble of all potential noisy environments. Intuitively, we also posit that both the IG and the KG contain certain noise-free subgraphs. Inspired by invariant learning literature [3, 7], we proceed to delineate formal definitions for these subgraphs.

ASSUMPTION 2.1 (INVARIANCE PROPERTY). *Consider a user-item bipartite graph $\mathcal{G}_b$ and a knowledge graph $\mathcal{G}_k$, there exist a noise-free bipartite subgraph $\mathcal{G}_b^I = \{\mathcal{V}_b, \mathcal{E}_b^I\}$ such that $\mathcal{E}_b^I \subseteq \mathcal{E}_b$, and task-relevant knowledge subgraph $\mathcal{G}_k^I \subseteq \mathcal{G}_k$. These subgraphs satisfy the following conditions: i) Suffciency condition: $\mathbf{y}_{ui} = f(\mathbf{u}, \mathbf{i}|\mathcal{G}_b^I, \mathcal{G}_k^I) + \epsilon$, where $\epsilon$ signifies an independent noise. ii) Invariance condition: $\forall s, s' \in \mathcal{S}, p_s(\mathbf{y}_{ui}|\mathbf{u}, \mathbf{i}, \mathcal{G}_b^I, \mathcal{G}_k^I) = p_{s'}(\mathbf{y}_{ui}|(\mathbf{u}, \mathbf{i}, \mathcal{G}_b^I, \mathcal{G}_k^I)$, where $p_s$ and $p_{s'}$ denote the distribution under environment $s$ and $s'$, respectively.*

It posits that these invariant subgraphs embedded within the IG and KG encapsulate sufficient information to forecast each user's interaction behaviors. Concurrently, their association with $\mathbf{y}_{ui}$ remains stable and invariant across environments characterized by varying noise distributions. This inherent invariance quality facilitates the effective filtering out of noise, thereby fostering the development of a robust and accurate recommendation model.

## 3 METHODOLOGY

In this section, we present the Knowledge Graph Invariant Learning (KGIL) framework for knowledge-aware recommendation. Drawing inspiration from invariant learning [3, 7, 28], KGIL endeavors to extract task-relevant knowledge connections from KG and highlight informative interactions from IG. The overview of KGIL is depicted in Figure 2. Now we illustrate the details of each component.

### 3.1 Environment Generator

KGs supplement items with auxiliary information, effectively mitigating the challenge of data sparsity in user-item interactions. However, numerous studies [13, 34] highlight that the construction

of KGs — typically decoupled from the user-item IG data collection — often leads to the inclusion of a vast array of task-irrelevant knowledge, which will detrimentally impact the recommendation performance. Therefore, to highlight task-relevant connections within the KG, we design $M$ independent environment generators to generate multiple KGs in diverse noisy environments. Specifically, given the KG data $\mathcal{G}_k = \{(h, r, t)|h, t \in \mathcal{V}_k, r \in \mathcal{E}_k\}$, each triplet $(h, r, t)$ will be associated with a random variable $p_m \sim$ Bernoulli$(\omega_m)$, where the triplet exists if $p_m = 1$ and is dropped otherwise. We parameterize the Bernoulli distribution weight $\omega_m$ via an MLP network $\Phi_m$:

$$\omega_m = \Phi_m(\mathbf{W}_r \mathbf{e}_h || \mathbf{e}_r || \mathbf{W}_r \mathbf{e}_t)), \quad (1)$$

where $||$ denotes the concat operation; $\mathbf{e}_h, \mathbf{e}_t, \mathbf{e}_r \in \mathbb{R}^d$ are the embedding for $h, t$ and $r$, respectively; $\mathbf{W}_r \in \mathbb{R}^{d \times d}$ is the transformation matrix of relation $r$, which projects entities from the $d$-dimension entity space into the relation space [37]. We use $\theta_m$ to summarize the parameters of the $m$-th environment generator. To train the model in an end-to-end manner, we relax the discrete $p_m$ to be a continuous variable in $[0, 1]$ and utilize the Gumbel-Max reparametrization trick. Specifically, we define the following function:

$$p_m = \text{sigmoid}((\log \delta - \log(1 - \delta) + \omega_m)/\tau), \delta \sim \text{Uniform}(0, 1), \quad (2)$$

where $\delta$ is a random variable sampled from a uniform distribution, and $\tau$ is the temperature hyperparameter. As the temperature $\tau \rightarrow 0$, $p_m$ gets close to binary. Hence, we can define $M$ independent environment generators to generate multiple new KGs $\{\mathcal{G}_k^1, ..., \mathcal{G}_k^M\}$ to simulate $M$ different environments.

### 3.2 Invariant Learning on Knowledge Graph

In this module, we aim to achieve task-relevant information extraction in multiple KGs by designing a shared invariant attention aggregation mechanism for KG data from diverse environments, and a knowledge-aware invariant learning task.

#### 3.2.1 *Relation-aware Invariant Attention Generation.* To automatically extract semantic information crucial for recommendation tasks from KG data, we turn to the principles of invariant learning [3, 7, 20, 29, 46]. This idea focuses on discovering invariant KG subgraphs across diverse environments. Recognizing that

KG subgraphs across different environments might have differing levels of noise, we find it imperative to design a feature learner that identifies these invariant subgraphs. We introduce the following attention score generation function for relation:

$$g(h, r) = \frac{\mathbf{e}_h \mathbf{W}_q \cdot (\mathbf{e}_r \mathbf{W}_k)^\top}{\sqrt{d}}, \tag{3}$$

where $\mathbf{W}_q, \mathbf{W}_k \in \mathbb{R}^{d \times d}$ are trainable weight matrices. This function determines the degree to which relations serve as the foundation for collaborative interactions for recommendation. For a given head $h$, we use softmax function to normalize the attention scores of its surrounding relations:

$$\hat{g}(h, r) = \frac{\exp(g(h, r))}{\sum_{(h, r') \in \mathcal{N}_h^r} \exp(g(h, r'))}, \tag{4}$$

where $\mathcal{N}_h^r$ refers to the set of neighbor relations centered on the head $h$. In order to further determine the importance of each knowledge triplet, we design the following attention score for knowledge triplets based on the attention score of relation:

$$\alpha(h, r, t) = \frac{\exp\left((\mathbf{e}_h || \hat{\mathbf{e}}_r) \cdot (\mathbf{e}_t || \hat{\mathbf{e}}_r)^\top\right)}{\sum_{(h, r', t') \in \mathcal{N}_h} \exp\left((\mathbf{e}_h || \hat{\mathbf{e}}_{r'}) \cdot (\mathbf{e}_{t'} || \hat{\mathbf{e}}_{r'})^\top\right)}, \tag{5}$$

where $\hat{\mathbf{e}}_r = \hat{g}(h, r)\mathbf{e}_r$; $\mathcal{N}_h$ refers to the set of neighbor triplets centered on the head $h$. We can observe that the design of $\alpha(h, r, t)$ first considers the attention score of the neighbor relations, and then considers the global information of the entire triplet. This hierarchical strategy can effectively measure the importance of each knowledge triplet in KG.

3.2.2 **Invariant Knowledge Aggregation**. In addition to designing the attention generation mechanism, following [39, 48], we also employ a relational-path aware aggregation mechanism for KG to distill the invariant knowledge subgraph. Specifically, for the given KG, we recursively learn the item and attribute representations in the following way:

$$\mathbf{e}_h^{(l+1)} = \frac{1}{|\mathcal{N}_h|} \sum_{(h, r, t) \in \mathcal{N}_h} \alpha(h, r, t) \mathbf{e}_r \odot \mathbf{e}_t^{(l)}, \tag{6}$$

where $\mathbf{e}_h^{(l+1)}$ and $\mathbf{e}_t^{(l+1)}$ denote the representations of head and tail, which memorize the relational signals propagated from their $l$-th layer neighbors. For each triplet $(h, r, t)$, a relational message $\mathbf{e}_r \odot \mathbf{e}_t^{(l)}$ is designed for implying different meanings of triplets, via modeling the relation $r$ through the projection or rotation operator [30]. Furthermore, we utilize the proposed attention score $\alpha(h, r, t)$ to decide how important each triplet is to the currently aggregated information. In subsequent designs, we will use invariant learning to constrain the attention mechanism to capture invariant KG subgraphs in diverse environments.

3.2.3 **Knowledge-aware Invariant Learning**. For the given KG data $\{\mathcal{G}_k^1, ..., \mathcal{G}_k^M\}$, we need to design the optimization objective to capture the invariant KG subgraph. Based on Assumption 2.1, the invariance condition states that the KG invariant subgraph should ensure invariant predictions across different environments. It emphasizes that the item representations enhanced by KG should be also invariant under different environments. Hence, we implement cross-environment invariance constraints through contrastive

learning. Specifically, given KG data $\mathcal{G}_k^m$ and $\mathcal{G}_k^{m+1}$, we aggregate $L$ times based on equation (6), and sum the item representations of these $L$ layers:

$$\mathbf{z}_i^m = \mathbf{e}_{m,i}^{(0)} + \cdots + \mathbf{e}_{m,i}^{(L)}, \qquad \mathbf{z}_i^{m+1} = \mathbf{e}_{m+1,i}^{(0)} + \cdots + \mathbf{e}_{m+1,i}^{(L)}, \tag{7}$$

where $\mathbf{e}_{m,i}^{(l)}$ and $\mathbf{e}_{m+1,i}^{(l)}$ denote the $l$-th layer item representations from $\mathcal{G}_k^m$ and $\mathcal{G}_k^{m+1}$, respectively. Then we adopt the projection head to map these representations into the space where invariant learning loss is calculated:

$$\hat{\mathbf{z}}_i^m = \hat{\mathbf{W}}^{(2)} \sigma(\hat{\mathbf{W}}^{(1)} \mathbf{z}_i^m + \mathbf{b}^{(1)}) + \mathbf{b}^{(2)},$$
$$\hat{\mathbf{z}}_i^{m+1} = \hat{\mathbf{W}}^{(2)} \sigma(\hat{\mathbf{W}}^{(1)} \mathbf{z}_i^{m+1} + \mathbf{b}^{(1)}) + \mathbf{b}^{(2)}, \tag{8}$$

where $\mathbf{W}$ and $\mathbf{b}$ denote the learnable weights and bias. To encourage the invariance, we should also define the positive and negative samples. For the given item $i$ in environment $\mathcal{G}_k^m$, we define positive samples as the same item $i$ in environment $\mathcal{G}_k^{m+1}$, and define negative samples as another different item $j \neq i$ in $\mathcal{G}_k^m$. Based on the positive and negative samples defined above, we define the following invariant learning optimization objective:

$$\mathcal{L}_{k,inv}^m = \frac{1}{|\mathcal{V}_i|} \sum_{i \in \mathcal{V}_i} -\log \frac{\exp(\text{sim}(\hat{\mathbf{z}}_i^m, \hat{\mathbf{z}}_i^{m+1})/\tau_k)}{\sum_{j \in \mathcal{V}_i, j \neq i} \exp(\text{sim}(\hat{\mathbf{z}}_i^m, \hat{\mathbf{z}}_j^m)/\tau_k)}, \tag{9}$$

where $\text{sim}(\cdot)$ denotes the similarity measurement, which is defined as the cosine similarity, and $\tau_k$ is the temperature hyperparameter. Please note that here we define $\mathcal{L}_m$ as an invariant learning objective between environment $m$ and $m + 1$. To fully explore the generated environments, we will consider $M$ losses in Section 3.4.

## 3.3 Invariant Learning on Interaction Graph

Equation (9) only considers the invariance constraints on the item representation space. According to the invariance condition in Assumption 2.1, we also need to establish a cross-environment invariant relationship for the interaction labels of the downstream recommendation tasks. The sufficiency condition requires that the captured invariant subgraphs can also correctly predict the interactions between users and items. Therefore, we need to combine the generated KGs with IG, and design a new aggregation mechanism for IG for downstream recommendation tasks, so as to achieve joint invariant learning of KG and IG.

3.3.1 **Attention-based Aggregation**. We hope to retain interactive edges that clearly reflect user interests and can better guide invariant learning of knowledge graphs. Therefore, we designed the following attention mechanism to highlight the important user-item interaction edges:

$$\beta(u, i) = \frac{\exp(LeakyReLU(\mathbf{W}_b[\mathbf{x}_i || \mathbf{x}_u]))}{\sum_{i \in \mathcal{N}_u} \exp(LeakyReLU(\mathbf{W}_b[\mathbf{x}_i || \mathbf{x}_u]))}, \tag{10}$$

where $\mathbf{x}_i$ and $\mathbf{x}_u$ represent the embedding of items and users, respectively; $\mathcal{N}_u$ represents the set of neighbor items of user $u$; and $\mathbf{W}_b$ is a learnable weight matrix. $LeakyReLU$ activation function is adopted for non-linear transformation. On the IG data, we design the following aggregation mechanism to update the embedding of users and items at each layer:

$$\mathbf{x}_u^{(l+1)} = \frac{1}{|\mathcal{N}_u|} \sum_{i \in \mathcal{N}_u} \beta(u,i) \mathbf{x}_i^{(l)}, \mathbf{x}_i^{(l+1)} = \frac{1}{|\mathcal{N}_i'|} \sum_{i \in \mathcal{N}_i'} \mathbf{x}_u^{(l)} + \mathbf{e}_i^{(l+1)}, \quad (11)$$

where $\mathcal{N}_i'$ represents the set of neighbor users of item $i$ on the IG graph, and $\mathbf{e}_i^{(l+1)}$ represents the $(l+1)$-th layer representation obtained by KG aggregation as shown in equation (6). Therefore, this aggregation method allows users and items to simultaneously consider the captured task-relevant semantic knowledge, thereby realizing KG-enhanced information aggregation.

*3.3.2* **Invariant Attention Learning**. In order to achieve invariance conditions in IG, we need to combine IG with KGs in different environments. Across these $M$ environments, we leverage the shared weight parameters to simultaneously aggregate information and learn invariant attention scores to KG and IG based on equations (6) and (11). Specifically, under environment $m$ and $m+1$, we aggregate $L$ times and sum the representations of these $L$ layers to obtain the final user and item representations:

$$\mathbf{h}_n^m = \mathbf{x}_{m,n}^{(0)} + \cdots + \mathbf{x}_{m,n}^{(L)}, \qquad \mathbf{h}_n^{m+1} = \mathbf{x}_{m+1,n}^{(0)} + \cdots + \mathbf{x}_{m+1,n}^{(L)}, \quad (12)$$

where the subscript $n$ indicates that we do not distinguish between users or items; $\mathbf{x}_{m,n}^{(l)}$ and $\mathbf{x}_{m+1,n}^{(l)}$ denote the $l$-th layer representations of from $m$ and $m+1$ environments, respectively. Similar to the invariant learning in KGs, we also learn the representations $\tilde{\mathbf{z}}_n^m$ and $\tilde{\mathbf{z}}_n^{m+1}$ through a project head. Finally, we define the following invariant learning objective in IG:

$$\mathcal{L}_{b,inv}^m = \frac{1}{|\mathcal{V}_b|} \sum_{n \in \mathcal{V}_b} -\log \frac{\exp(\text{sim}(\tilde{\mathbf{h}}_n^m, \tilde{\mathbf{h}}_n^{m+1})/\tau_b)}{\sum_{j \in \mathcal{V}_b, j \neq n} \exp(\text{sim}(\tilde{\mathbf{h}}_n^m, \tilde{\mathbf{h}}_j^m)/\tau_b)}, \quad (13)$$

where $\tau_b$ is the temperature hyperparameter. Equation (13) is defined from the perspective of the global view, which considers both KG and IG information, thus achieving knowledge-enhanced invariant learning for user and item representations.

*3.3.3* **Invariant Model Prediction**. We implement the sufficiency condition in Assumption 2.1 by defining invariant model predictions under $M$ different environments. For item and user representations in environment $m$, we predict their matching scores through the inner product, and then we define the BPR loss as follows:

$$\mathcal{L}_{bpr}^m = \sum_{(u,i,j) \in \mathcal{D}} -\log \sigma(\hat{y}_{ui}^m - \hat{y}_{uj}), \quad \hat{y}_{ui}^m = \mathbf{h}_i^{m\top} \mathbf{h}_i^m, \quad (14)$$

where $\mathcal{D} = \{(u,i,j)|(u,i) \in \mathcal{D}^+, (u,j) \in \mathcal{D}^-\}$ is the training dataset consisting of the observed interactions $\mathcal{D}^+$ and unobserved counterparts $\mathcal{D}^-$; $\sigma$ is the sigmoid function. Under environment $m$, we sum the invariant learning in KG and global invariant learning in IG and define them as $\mathcal{L}_{inv}^m = \mathcal{L}_{k,inv}^m + \mathcal{L}_{b,inv}^m$. For simplicity, we also define the optimization goal of invariant model prediction as: $\mathcal{L}_{rec}^m = \mathcal{L}_{bpr}^m + \mathcal{L}_{inv}^m$. In the inference stage, we adopt the mean representations of the items and users in $M$ environments to make the final predictions.

## 3.4 Model Optimization

*3.4.1* **Cross-environment Invariant Learning**. So far, we have used environment $m$ as an example to achieve the sufficiency and

invariance conditions of invariant learning. In order to consider the invariance principle across more diverse environments, for the given $M$ environment generators, we achieve our final optimization goal by minimizing the mean and variance over $M$ environments:

$$\Theta^* = \arg\min_{\Theta} \left\{ \frac{1}{M} \sum_{m=1}^M \mathcal{L}_{rec}^m + \lambda \text{Var}(\{\mathcal{L}_{inv}^m : 1 \leq m \leq M\}) \right\}, \quad (15)$$

where $\Theta$ outlines all the parameters that can be learned in the model except the environment generators; $\lambda$ is a hyperparameter. This way of achieving sufficiency and invariance under different environments allows the model to accurately capture task-relevant KG subgraphs and informative IG subgraphs, thereby achieving more accurate and robust recommendation predictions.

*3.4.2* **Multi-environment Exploration**. The purpose of the environment generators is to generate more diverse noisy environments, thereby improving the robustness of the model. To fully explore the types of noise that may be ignored for the current given model, we update the parameters of the environment generator by defining the following optimization goals:

$$\theta_1^*, ..., \theta_M^* = \arg\max_{\theta_1,...,\theta_M} \text{Var}(\{\mathcal{L}_{inv}^m : 1 \leq m \leq M\}). \quad (16)$$

It encourages the environment generators to explore more diverse environments by maximizing the variance of $M$ invariant learning objectives, which can further generate environments that are challenging for the model, thus improving the robustness against diverse noise. Combining equations (15) and (16), we find that it is a bi-level optimization problem, which is usually difficult to solve. In our implementation, we use an alternating optimization strategy to update the model parameters and environment generator parameters separately. Specifically, we define $T$ iterations as a cycle. During the process of model training, the parameters of environment generators will be updated every $T$ iterations.

## 3.5 Model Analysis

*3.5.1* **Model Size**. The model parameters of KGIL include: 1) $M$ environment generators $\{\theta_1, ..., \theta_M\}$; 2) ID embeddings of users, items, relations, and attribute entities $\{\mathbf{e}_u, \mathbf{e}_i, \mathbf{e}_r, \mathbf{e}_e | u \in \mathcal{U}, i \in \mathcal{I}, r \in \mathcal{E}_k, e \in \mathcal{V}_k\}$; and 3) Weights for attention generation (*i.e.*, $\mathbf{W}_q, \mathbf{W}_k, \mathbf{W}_b$), and project heads for invariant learning.

*3.5.2* **Time Complexity**. While KGIL employs $M$ distinct environment generators, we contend that its time complexity remains on par with prevailing methods [48, 49, 53, 54]. Our time complexity is primarily derived from three main components: 1) Environment generators have a complexity of $O(M|\mathcal{V}_k|d)$ to generate KGs. As we update the parameters of generators periodically (*i.e.*, every $T$ iterations), the computational overhead is mitigated. 2) Aggregation of KG and IG demands $O(M(|\mathcal{V}_k| + |\mathcal{V}_b|)dL)$. 3) Invariant learning has a complexity of $O(MB(|\mathcal{U}| + 2|\mathcal{I}|)d)$, with $B$ representing the distinct count of users and items within a batch. It's notable that most current efforts [48, 54] utilize two or more views for contrastive learning. Consequently, their aggregation process typically demands multiple times of $O((|\mathcal{V}_k| + |\mathcal{V}_b|)dL)$. In our implementation, we observed optimal performance when setting $M = 3$, indicating that our environment generators do not add significant complexity. Considering the performance and complexity

**Table 1: Statistics of experimental datasets.**

| Statistics | Last-FM | Yelp2018 | MIND |
|---|---|---|---|
| # Users | 23,566 | 45,919 | 100,000 |
| # Items | 48,123 | 45,538 | 30,577 |
| # Interactions | 3,034,796 | 1,183,610 | 2,975,319 |
| # Density | 2.7e-3 | 5.7e-4 | 9.7e-4 |
| Knowledge Graph | | | |
| # Entities | 58,266 | 47,472 | 24,733 |
| # Relations | 9 | 42 | 512 |
| # Triplets | 464,567 | 869,603 | 148,568 |

trade-offs, we believe that the marginal complexity introduced by our technique is justifiable.

## 4 EXPERIMENTS

To evaluate the effectiveness of KGIL, we carry out a series of experiments to address the following **R**esearch **Q**uestions:

- **RQ1:** How does KGIL compare against various state-of-the-art recommendation models?
- **RQ2:** What is the contribution of each component within the KGIL framework to its overall recommendation performance?
- **RQ3:** Can KGIL aid the model in acquiring robust representations and how proficient is it at mitigating noise issues?

### 4.1 Experimental Settings

*4.1.1 Datasets.* To ensure a comprehensive and persuasive evaluation, we utilize three publicly available recommendation datasets, each reflecting distinct real-world scenarios:

- **Last-FM:** Originating from user listening records on music platforms, it treats various tracks as candidate item sets.
- **Yelp2018:** Derived from the 2018 edition of Yelp's rating data, it focuses on local businesses, which are designated as items.
- **MIND:** Sourced from the anonymous behavior logs on the Microsoft News website, it includes user clicks on news articles and knowledge graphs built from Wikidata using headline entities.

These datasets exemplify music, businesses, and news recommendation scenarios, respectively. Regarding interaction data, we employ the 10-core preprocessing method to refine the datasets, retaining only users and items with occurrences exceeding 10 times. For the knowledge graph data, we follow existing literature and adopt adaptive construction techniques tailored for each dataset. The detailed statistics of these datasets are shown in Table 1.

*4.1.2 Evaluation Metrics.* To maintain a consistent comparison, we adopt the full-ranking strategy across all experiments when assessing recommendation performance. For each user, items with which they have never interacted are considered negative, while items included in the dataset are deemed positive. For top-N recommendations, we rely on two widely-used evaluation metrics: Recall@N and NDCG@N, with N defaulting to 20. Ultimately, we present the average outcomes for all users within the testing set.

*4.1.3 Baselines and Settings.* To demonstrate the superiority of our proposed method, we incorporate a variety of baseline methods. These span from traditional CF models to the embedding-based

**Table 2: Overall performance comparison for all methods on three datasets. The highest results are in bold.**

| Method | Last-FM | | Yelp2018 | | MIND | |
|---|---|---|---|---|---|---|
| | Recall | NDCG | Recall | NDCG | Recall | NDCG |
| BPRMF | 0.0715 | 0.0637 | 0.0625 | 0.0388 | 0.0384 | 0.0253 |
| NeuMF | 0.0699 | 0.0615 | 0.0631 | 0.0390 | 0.0308 | 0.0237 |
| GC-MC | 0.0709 | 0.0631 | 0.0659 | 0.0410 | 0.0386 | 0.0261 |
| LightGCN | 0.0738 | 0.0647 | 0.0661 | 0.0415 | 0.0408 | 0.0266 |
| SGL | 0.0879 | 0.0775 | 0.0684 | 0.0431 | 0.0416 | 0.0272 |
| CKE | 0.0732 | 0.0630 | 0.0651 | 0.0414 | 0.0387 | 0.0247 |
| KTUP | 0.0783 | 0.0681 | 0.0640 | 0.0420 | 0.0362 | 0.0302 |
| RippleNet | 0.0791 | 0.0684 | 0.0664 | 0.0428 | 0.0372 | 0.0283 |
| CKAN | 0.0812 | 0.0660 | 0.0622 | 0.0389 | 0.0361 | 0.0275 |
| KGNN-LS | 0.0880 | 0.0642 | 0.0637 | 0.0402 | 0.0395 | 0.0302 |
| KGAT | 0.0873 | 0.0744 | 0.0712 | 0.0443 | 0.0340 | 0.0287 |
| KGIN | 0.0900 | 0.0779 | 0.0736 | 0.0482 | 0.0380 | 0.0293 |
| MCCLK | 0.0671 | 0.0603 | 0.0630 | 0.0397 | 0.0327 | 0.0194 |
| KGCL | 0.0896 | 0.0806 | 0.0738 | 0.0487 | 0.0351 | 0.0221 |
| KGRec | 0.0936 | 0.0805 | 0.0741 | 0.0482 | **0.0419** | 0.0306 |
| KGIL | **0.0942** | **0.0817** | **0.0745** | **0.0498** | 0.0417 | **0.0310** |

method, GNN-based knowledge-aware recommenders, and self-supervised knowledge-graph recommendation methods. Owing to spatial constraints, comprehensive descriptions of these baseline methods and settings can be found in Appendix A.

### 4.2 Performance Comparison (RQ1)

In evaluating recommendation performance, we compare our proposed KGIL with existing baseline methods across three datasets, yielding the subsequent observations:

- Most KG-enhanced recommendation models significantly surpass the traditional CF-based methods. Nevertheless, a discernible performance gap remains between some KG-enhanced techniques and the state-of-the-art self-supervised learning method, *e.g.,* SGL, in the CF domain. This discrepancy may stem from the inherent nature of most existing KG data, which is not specifically tailored for recommendation tasks, thereby incorporating vast amounts of task-irrelevant connections. If the recommendation systems indiscriminately assimilate the information without discerning the crucial knowledge, it will involve substantial noise for the representation learning of users and items. Contrarily, our method effectively identifies and leverages invariant attentive subgraphs from KGs via the proposed invariant attention generation mechanism. It ensures a holistic and precise utilization of the task-relevant triplet data within KGs, potentially paving the way to overcome these limitations.
- When set against self-supervised learning endeavors, such as SGL, MCCLK, and KGCL, our method manifests significant improvements. A closer examination reveals that while most current self-supervised learning techniques rely on random data augmentation to generate views, KGIL implements learnable environment generators for this purpose. These environment generators explore a broader spectrum of noisy environments by adversarially maximizing the variance in invariant learning objectives. Concurrently, our design encourages the model to attenuate both the mean and variance of these objectives. It

**Table 3: The impact of different components in KGIL.**

| Ablation | MIND | | Last-FM | | Yelp2018 | |
|---|---|---|---|---|---|---|
| | Recall | NDCG | Recall | NDCG | Recall | NDCG |
| KGIL | **0.0417** | **0.0310** | **0.0942** | **0.0817** | **0.0745** | **0.0498** |
| w/o EG | 0.0401 | 0.0296 | 0.0923 | 0.0795 | 0.0726 | 0.0473 |
| w/o IKG | 0.0385 | 0.0282 | 0.0912 | 0.0789 | 0.0719 | 0.0466 |
| w/o IIG | 0.0399 | 0.0291 | 0.0930 | 0.0801 | 0.0740 | 0.0494 |

endows our model with an enhanced capability to filter out the noise and pinpoint task-relevant information.

## 4.3 Ablation Study (RQ2)

### 4.3.1 *Impact of Different Components.* To assess the individual contributions of KGIL's components to its performance, we evaluate the following three model variants:

- "w/o EG": This version excludes the environment generators in KGIL. In its place, we employ random data augmentation to simulate diverse KG environments.
- "w/o IKG": This KGIL variant omits the invariant attention mechanism for KG. Consequently, all triplet information is aggregated with uniform attention scores.
- "w/o IIG": Without the invariant attention mechanism for the user-item interaction graph in this KGIL variant, users and items propagate and aggregate messages across interaction graph connections with consistent attention scores.

As depicted in Table 3, KGIL's performance diminishes to varying extents when any of these components is absent. The "w/o EG" variant exhibits the most pronounced degradation. This aligns with our assertion that learnable data augmentation—by enhancing the spectrum of noisy environments—empowers the model to discern task-specific details and cultivate robust, invariant representations. Concurrently, the attention mechanism not only enables the model to filter irrelevant information but also to harness triplet facts of varying significance, thereby optimizing information utilization and bolstering recommendation performance.

### 4.3.2 *Sensitivity to the key hyperparameters.* We further investigate the influence of two pivotal hyperparameters in KGIL: $M$ and $\lambda$. Here, $M$ denotes the number of environments, and $\lambda$ signifies the coefficient regulating the variance. We establish their value ranges as $M \in \{2, 3, 4, 5\}$ and $\lambda \in \{0.1, 0.5, 1.0, 1.5, 2.0\}$, with the corresponding outcomes illustrated in Figure 3. From the results, performance is generally improved when employing more than three environments. As for the variance-controlling coefficient $\lambda$, performance variations are scarcely noticeable once the coefficient surpasses 0.5. Consequently, the final performance of the model showcases resilience against variations in these hyperparameters.

## 4.4 Model Benefits Analysis (RQ3)

### 4.4.1 *Robustness to Information Noise.* We conduct experiments to assess how KGIL handles noisy triplets in KG and interactions in IG. To do this, we introduce varying levels of random noise. In the KG, we randomly select a designated proportion of triplets and replace their tail entities. For the user-item interaction graph, we add a set proportion of random noisy interaction edges to the

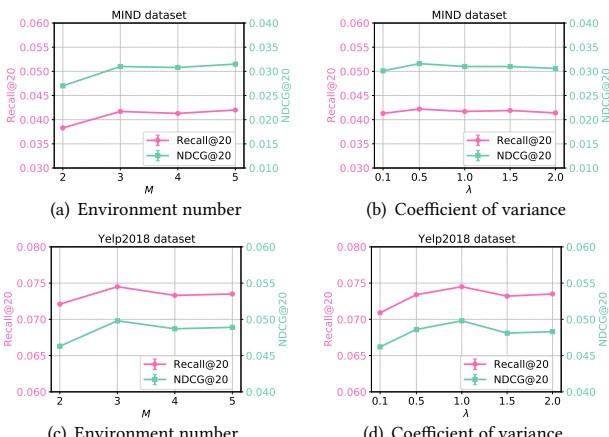

(a) Environment number     (b) Coefficient of variance

(c) Environment number     (d) Coefficient of variance

**Figure 3: Performance over diverse hyperparameters.**

training data, while the test set remains unchanged. Figure 4 shows KGIL's performance in comparison to established models such as SGL, KGIN, KGCL, and KGRec. We conduct experiments on the MIND and Yelp2018 datasets. While noise reduces the performance of all models, KGIL demonstrates a lesser degree of degradation.

In the experiment with noisy triplets, as the noise ratio increases, KGIL's performance consistently stays within an acceptable range. Its robustness surpasses that of other baseline models. This highlights KGIL's ability to generate diverse noisy environment distributions, enabling the model to extract robust and invariant representations from data, even amidst noise. In essence, unlike conventional models that mainly rely on data augmentation, KGIL employs a learnable data augmentation strategy. From an adversarial perspective, it also encourages the diversity of environments and the invariance of subgraph learning. Hence, it helps the model to filter out noise and discern the relevance of different triplet facts in KG according to the downstream recommendation task.

The experiment with noisy interactions in IG shows a more significant performance drop than that with noisy triplets in KG. We postulate that the intrinsic sparsity of interaction graph data exacerbates the impact of added noise interactions on recommendation efficacy, relative to noise introduced to triplets. Nevertheless, from multiple curve trends, the performance of KGIL consistently outperforms its competitors, with a rate of decline parallel to the SGL model. We attribute this robustness to KGIL's fusion of invariant learning and the attention mechanism for KG data. By leveraging the environmental diversity and the principle of invariance in KG learning, KGIL derives robust entity representations. Hence, it encourages the model to highlight task-relevant knowledge from noisy KG. Furthermore, the invariant attention-based aggregation process in both KG and IG, enhances the invariant representation learning for users and items with task-relevant knowledge, thereby bolstering the recommendation performance.

### 4.4.2 *Cold-Start Recommendation.* To evaluate the efficacy of our framework under a cold-start scenario characterized by extremely sparse interaction data, we categorize users in the dataset into five groups based on their interaction frequency. A lower group ID corresponds to decreased user activity, exacerbating the cold-start challenge. We compare KGIL with several baselines across

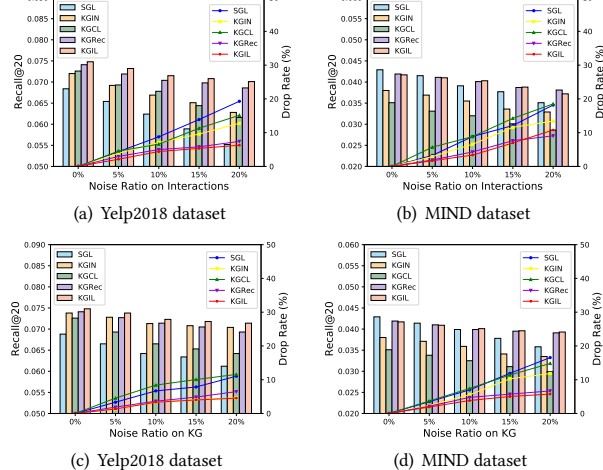

(a) Yelp2018 dataset

(b) MIND dataset

(c) Yelp2018 dataset

(d) MIND dataset

**Figure 4: Impact comparison *w.r.t.* noise ratio added in KG and IG. The bar displays Recall@20, while the curve displays the drop percentage of performance.**

varying cold-start levels. The results are depicted in Figure 5. The results clearly indicate that KGIL adeptly addresses the cold-start issue, consistently outperforming other models. The superior performance stems from KGIL's ability to derive robust item representations by pinpointing informative connections within the KG. Subsequently, it employs an invariant attention mechanism to utilize task-relevant information towards interaction graph learning. Hence, even in the case of cold-start users, KGIL equips the recommendation model to discern and reflect users' potential preferences with remarkable accuracy.

## 5 RELATED WORK

**Knowledge-aware Recommendation.** Due to the existence of extensive prior information, KG can effectively alleviate many problems in recommendation models, including data sparsity or cold start issues. Early studies [6, 31, 34, 51] are mainly based on embedding methods, which use embedding technology to represent the relations and entities in KG, and provide additional prior information for the recommendation model. Represented by CKE [51] and TransE [5], these methods [6, 47] tend to learn first-order graph connections. Despite their effectiveness, they still fall short in modeling long-range semantics and latent user preferences. The path-based efforts [16, 17, 33, 40] explore the long-range connectivity between target users and item entities by extracting different semantic paths via KG. For example, RippleNet [33] collects paths from users to historical items, and then injects the knowledge information from item entity representation into user representations. However, with the large-scale growth of interaction data, such methods [17, 33] require either extremely time-consuming search, or artificial pre-definition of domains to filter paths. Until recently, GNN-based methods [35–37, 39, 43] use the message-passing mechanism to absorb structural information into node representations, thereby achieving strong representation capabilities for users and items. CKAN [43] adopts different propagation strategies for knowledge signals and collaborative signals. KGAT [37] employs the attention mechanism to perform graph convolution operations on the

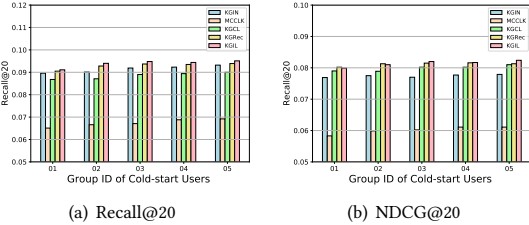

(a) Recall@20

(b) NDCG@20

**Figure 5: Recommendation performance over different cold-start user groups on Last-FM dataset.**

heterogeneous hybrid graph. KGIN [39] adopts an adaptive aggregation method on the heterogeneous graph and uses high-order information to learn fine-grained user potential intentions.

**Invariant Learning in Graphs and Recommendation.** Invariant learning [1, 3, 7, 8, 27] is emerging as a pivotal technique to enhance model generalization. This approach often posits that certain stable features exist within data that causally determine the target labels. Moreover, the relationship between these stable features and labels remains invariant across different environments. Conversely, environmental features tend to encapsulate information that lacks a causal linkage to labels [11, 46], often manifesting as shortcuts [12] or noise [29]. Recently, the graph field has seen a surge in the application of invariant learning [18, 24, 45]. To discern stable features, DIR [46] makes interventions on environmental features, whereas GREA [23] employs both environment removal and replacement data augmentation. Meanwhile, CAL [29] and DisC [11] leverage random replacement of environmental features to segregate stable features from environmental ones. Correspondingly, there have been initiatives to integrate the principles of invariant learning into recommendation systems. For instance, InvPref [42] iteratively decomposes the invariant preference and variant preference from biased observational user behaviors to achieve debiasing in the recommendation. InvCF [50] addresses the issue of popularity distribution shifts in CF models by acquiring invariant representations. Furthermore, invariant learning has showcased impressive outcomes in diverse recommendation contexts, including CTR prediction [52] and multimedia recommendations [9].

## 6 CONCLUSION

In this paper, we propose KGIL, a novel framework designed to enhance knowledge-aware recommendation systems. Drawing inspiration from invariant learning, our approach equips the KG-enhanced recommendation models with the capacity to capture invariant subgraphs across environments, thereby harnessing the full potential of the side information provided by the KG. To achieve this, we begin by generating the noisy environments from KG, facilitated by the design of multiple environment generators. Subsequently, we design attention mechanisms tailored for both KG and interaction graphs and encourage the model to extract cross-environment invariant attentive subgraphs. Finally, adversarial optimization is employed to encourage the diversity of generated environments and the invariance of the representation learning. We conduct extensive experiments and compare KGIL with existing efforts. Experimental results further demonstrate the superiority of our proposed KGIL framework.

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

# A  MORE IMPLEMENTATION DETAILS

## A.1  Baselines

- **BPRMF** [26] is a classic matrix factorization (MF) method that ranks candidate items based only on implicit feedback and calculates pairwise ranking loss with user and item ID embeddings.
- **NeuMF** [15] integrates multi-layer perceptrons into MF to learn nonlinear feature interactions between users and items.
- **GC-MC** [4] utilizes a bipartite graph to model recommendation tasks and captures interaction patterns in the form of link prediction based on the graph autoencoder framework.
- **LightGCN** [14] is a GNN-based model, which realizes message propagation between users and items by simplifying GCN.
- **SGL** [44] is a self-supervised learning method for GNN-based recommendation, it constructs multiple graph views for contrastive learning to obtain robust representations for CF.
- **CKE** [51] is an embedding-based method using TransR to guide the entity representation in KG, integrating the derived semantic information into the CF framework for enhancement.
- **KTUP** [6] lets the KG and interaction graph strengthen each other, so that the learned representation contains the relationship knowledge between entities that are complementary to the interaction.
- **RippleNet** [33] is an embedding-based approach that leverages ideas that propagate user preference over KG.

- **KGNN-LS** [35] takes into account the user's preference for KG relations and the label smoothing problem of information aggregation to generate a user-specific item representation.
- **KGAT** [37] combines the KG and interaction graphs and designs an attention mechanism-based messaging strategy for recursively propagating user/item embeddings.
- **CKAN** [43] based on KGNN-LS, adopts different neighborhood aggregation mechanisms for user-item interaction graph and KG respectively to obtain the embedding of users and items.
- **KGIN** [39] uses KG as auxiliary information to learn the user's potential intentions, and captures long-range semantic information through the relation-aware aggregation strategy.
- **MCCLK** [54] performs hierarchical contrastive learning on multi-level views while leveraging structural semantics and collaborative semantics to mine additional supervisory signals.
- **KGCL** [49] performs contrastive learning on KG, aiming to reduce the impact of information noise on the knowledge graph-enhanced recommendation system.
- **KRDN** [53] proposes a pruning-based denoising framework, which is employed on irrelevant knowledge connections and noisy interactions.
- **KGRec** [48] devises a self-supervised rationalization method to identify informative knowledge connections.

