# OpenReview forum: "Unleashing the Power of Knowledge Graph for Recommendation via Invariant Learning"
_ACM.org/TheWebConf/2024/Conference — TheWebConf24_

### Official Review · Reviewer_QAe7 · 2023-11-21

**Novelty:** 4
**Technical Quality:** 4

**Review:**

The authors propose a knowledge-enhanced recommendation model, which generates multiple noisy environments and simultaneously learns the environment-invariant information for a recommendation. The environment-invariant idea is new and interesting. The paper is also generally well-written.

Pros:
- The environment-invariant idea is new and interesting.
- The paper is also generally well-written.
- Extensive experiments are conducted.

Cons:
- The performance of the proposed method does not show significant improvements. For example, the improvements in the recall metric are less than 1% on all datasets.
- Some important baselines are missing. Knowledge-enhanced recommendation still has room for improvements, and it has been observed that knowledge-enhanced recommendation models may even underperform the existing contrastive learning models such as SimGCL [1]. The authors may need to compare with such baselines.

[1]Are graph augmentations necessary? simple graph contrastive learning for recommendation.SIGIR-2022

**Questions:**

- When optimizing the loss in Eq(16), how did the authors update the parameter in Eq(1)?

- How did the authors calculate the rating score for unobserved items in the inference stage? Is it still based on Eq(14)?  It seems unnecessary to generate noisy environments during the inference stage, as unstable results may be obtained.

- There is a typo in Eq(14).

**Reviewer Confidence:**

3: The reviewer is confident but not certain that the evaluation is correct

**Scope:**

3: The work is somewhat relevant to the Web and to the track, and is of narrow interest to a sub-community

---

### Official Review · Reviewer_u9aV · 2023-11-21

**Novelty:** 4
**Technical Quality:** 3

**Review:**

This article employs a generator to create several knowledge graphs and user-item graphs with noise. It designs a network based on attention to encode representations of users and items using these noisy graphs. Furthermore, it utilizes contrastive learning to enhance noise-independent aspects of the representations, thereby learning invariant representations. Additionally, adversarial learning is employed to strengthen the generator, aiming for improved learning outcomes.

Pros:
1. The motivation of the article is reasonable, addressing the noise-rich nature of knowledge graphs in the context of recommendation is beneficial for improving performance.
2. Combining contrastive learning and adversarial learning to learn invariant representations makes sense.
3. The experiments seems to show the effectiveness of the proposed work.

Cons:
1. Some experiments related to the motivation of KGIL are absent.
2. Performance improvement is not significant.
3. Lacks some theoretical support.

**Questions:**

Q1: The author mentions the sparsity issue in the Introduction, but there are no experiments related to this issue in the paper, only the noisy experiment. Does the model also perform well compared to different baselines under various sparsity ratios (like randomly removing a portion of interactions)?

Q2: Is there any theoretical support for invariant learning with contrastive learning and adversarial learning?

Q3: In Figures 4(b)-(d), it seems that for the MIND dataset, without noise, SGL performs better than KGIL, but in Yelp2018, it is not the case. Why does this happen? Why does KGIL not perform well in the clean MIND dataset?

**Reviewer Confidence:**

3: The reviewer is confident but not certain that the evaluation is correct

**Scope:**

4: The work is relevant to the Web and to the track, and is of broad interest to the community

---

### Official Review · Reviewer_wtJp · 2023-11-21

**Novelty:** 5
**Technical Quality:** 5

**Review:**

**Summary**

The paper proposes a novel framework, namely Knowledge Graph Invariant Learning (KGIL), to accomplish the recommendation task by integrating information from multiple knowledge graphs. To make the most of the multi-source knowledge graph information while ensuring the effectiveness of task-specific information extracted in downstream tasks and various scenarios, the author team designed an independent environment generator for different scenarios to simulate potential noise and extract task-specific connections in KG. In addition, attention-based graph learning is designed and applied to the extracted KG and the original Interaction Graph to extract task-relevant information from multiple KGs and combine relevant information to achieve the recommendation task. To demonstrate the effectiveness of the proposed model, the authors tested it on three different real open-source datasets, and related ablation experiments also strongly confirmed the rationality of each module.

**Strengths**

1. The paper has rigorous problem definition and symbol usage, smooth writing, rich content, reasonable structure, clear and beautiful graphics, and high readability.
2. The motivation of the paper is reasonable. Based on the current situation of large information and noise in recommendation systems, a learning framework for integrating multiple knowledge graphs for different scenarios is proposed, which meets the needs of the industry.
3. The design of the model components is novel, can be completed with relatively simple components, and has strong pertinence and feasibility, which can naturally achieve KG task-related information extraction and IG related information fusion.
4. The model analysis is complete, with detailed analysis of parameter quantity and time complexity of each module. Even in practical industrial scenarios, the complexity of the model is acceptable.

**Weaknesses**

The technical originality in the proposed model is relatively low, and the techniques used in the method are relatively conventional. It appears that many of the model components and optimization methods are simply a combination of existing methods, resulting in the proposed model being perceived as an incremental work.

**Questions:**

1. The definition of "noisy environments" in the knowledge graph and the corresponding "Environment Generator" component mentioned in the paper is relatively abstract. I wonder if the author team can provide a case study example to demonstrate the existence of the problem and further prove the effectiveness of the model.

2. I noticed that the optimization function of the model in the paper is simply the sum of three losses (the invariant learning in KG, global invariant learning in IG, and BPR loss). I would like to know if the authors have conducted parameter sensitivity analysis to further demonstrate the impact of these three mechanisms on the performance of the model.

3. In fact, knowledge graphs exist as a form of knowledge storage. I wonder can other forms of data, such as knowledge in LLMs, which are currently popular, also be used as supplementary data for models to achieve the same effect of KG learning from another perspective?

**Reviewer Confidence:**

3: The reviewer is confident but not certain that the evaluation is correct

**Scope:**

4: The work is relevant to the Web and to the track, and is of broad interest to the community

---

### Official Review · Reviewer_15PZ · 2023-11-23

**Novelty:** 5
**Technical Quality:** 5

**Review:**

In this paper, the author proposes Knowledge Graph Invariant Learning (KGIL), a novel framework aimed at enhancing knowledge-aware recommendation systems.

The motivation stems from the recognition that existing models often aggregate knowledge graph (KG) information indiscriminately, potentially introducing noisy knowledge that hampers recommendation quality. KGIL introduces the principle of invariance to knowledge-aware recommendation, focusing on discerning and leveraging task-relevant knowledge connections within the KG.
The KGIL framework begins by generating diverse noisy KG-environments using multiple environment generators. The next step involves the design of attention mechanisms tailored for both the KG and user-item interaction graph. These mechanisms aim to facilitate the learning of environment-invariant subgraphs, allowing the model to capture invariant knowledge across different environments. Adversarial optimization is then employed to enhance both the diversity of generated environments and the invariance of representation learning.

The proposed methodology outperforms state-of-the-art (SOTA) approaches, as evidenced by robust results confirmed through ablation studies, affirming its efficacy.
Noteworthy is the approach's ability to achieve superior performance with a minimal increase in model size and time complexity compared to other SOTA methods.

While the proposed approach is commendable, a notable concern lies in the absence of the provided code, raising concerns about the reproducibility of the results.


Equation 14 requires rectification, specifying $h_u^{mT} h_i^m$ instead of the erroneously stated $h_i^{mT} h_i^m$.
Additionally, in Equation 10, it is suggested that the authors acknowledge the origin of the equation by referencing Graph Attention Networks (GATs), as it closely resembles Equation 3 from GATs.

In Section 3.3, where the authors mention using mean representations of items and users in M environments during the inference stage, it is recommended to include the corresponding equation for clarity, though it is acknowledged that the statement is comprehensible as is.

Regarding Section 4.3.1, where the authors mention employing random data augmentation to simulate diverse KG environments, a more detailed explanation of their methodology is warranted.

The term "data augmentation" may be misleading, as it suggests the addition and/or removal of random triples, implying random edges between nodes.

However, the environmental generator's mechanism, described in section 3.1, involves only the removal of edges, as expressed by the authors: "each triple will be associated with a random variable p_m, where the triplet exists if $p_m = 1$ and is dropped otherwise."
As a result, the ablation study should specifically involve randomly removing  (and not adding) certain edges between nodes (edge dropout). The current description lacks clarity on this point, creating uncertainty about the exact methodology employed.

Moreover, the documentation of the experiments is deficient, with numerous crucial details regarding the experimental setup conspicuously absent. Key parameters such as embedding size, matrix dimensions, and the number of layers in the graph neural network-based recommender system are notably absent from the discussion.
Despite the author's assertion in Section 4.1.3 that a comprehensive description of settings is available in Appendix A, a closer examination reveals that these essential details are also omitted from the appendix.

In summary, my overall positive assessment of the work and appreciation for the proposed method are negatively influenced by concerns related to reproducibility.
I kindly recommend that the authors provide the code of their work and a thorough description of the experimental setup. This would greatly enhance the transparency and reproducibility of the research.

*Pros*:
- Novel approach that surpasses SOTA methods with only a marginal increase in model size and time performance.
- The paper is well-crafted, demonstrating clarity and a coherent storyline.
- Adequate empirical analyses and discussions effectively substantiate the motivations and claims presented in the paper.

*Cons*:
- Absence of provided code.
- Minor errors detected, e.g., in equation 14.
- Certain concerns raised regarding the ablation study; please refer to the full review for details.
- Incomplete documentation of the experimental setup.

**Questions:**

- Can the authors provide a more detailed explanation of their methodology in Section 4.3.1, specifically concerning the use of “random data augmentation” to simulate diverse KG environments?
- Could the authors provide a more detailed explanation for the statement in Section 4.1.1, which mentions, "For the knowledge graph data, we follow existing literature and adopt adaptive construction techniques tailored for each dataset"? The current description appears somewhat vague, and additional elaboration on the specific adaptive construction techniques used for individual datasets would be appreciated.

**Ethics Review Description:**

No issue

**Reviewer Confidence:**

4: The reviewer is certain that the evaluation is correct and very familiar with the relevant literature

**Scope:**

3: The work is somewhat relevant to the Web and to the track, and is of narrow interest to a sub-community

---

### Official Review · Reviewer_mPE4 · 2023-11-30

**Novelty:** 3
**Technical Quality:** 4

**Review:**

This paper argued that most existing knowledge-aware recommendation models indiscriminately aggregate all information in KG, and could introduce additional noisy knowledge into representation learning. So they proposed a Knowledge Graph Invariant Learning (KGIL) framework based the principle of invariance, which can simulate diverse noisy KG-environments. They also devised a novel attention mechanism to learn invariant representation across environments.Experiments on three datasets demonstrated the superiority of KGIL.
However, I must point out some issues below:
Line 120 in the introduction section，the path “𝑢1 →…→ 𝑖4, and the path 𝑢2 →…→ 𝑖3” does not match what above.
Overall, the article is well structured and motivated. Unfortunately, the paper lacks technical innovation, and is based on a simplistic methodology. It doesn't get to the root of the noise problem. Therefore, I think the paper has some room for improvement.

**Questions:**

The meaning of Figure 1(b) is not very clearly explained.
In 2.2, the parameters in the equation for the invariance condition are exactly the same, please explain.
The way to deal with knowledge graph noise in articles is to use attention mechanism, but this approach has been used in many articles for a long time.

**Reviewer Confidence:**

4: The reviewer is certain that the evaluation is correct and very familiar with the relevant literature

**Scope:**

4: The work is relevant to the Web and to the track, and is of broad interest to the community

---

### Decision · Program_Chairs · 2024-01-22

**Decision:**

Accept

**Comment:**

All reviewers found merits in the submission but some of them also propose a number of concerns. I think the authors have fixed many of these concerns in the rebuttal discussions and I believe they will improve their submission per the suggestions.